# PreferenceNet: Encoding Human Preferences in Auction Design with Deep Learning

**Neehar Peri\*, Michael J. Curry\*, Samuel Dooley, John P. Dickerson**
Center for Machine Learning, University of Maryland
peri@umiacs.umd.edu, {curry, sdooley1, john}@cs.umd.edu

## Abstract

The design of optimal auctions is a problem of interest in economics, game theory, and computer science. Despite decades of effort, strategyproof, revenue-maximizing auction designs are still not known outside of restricted settings. However, recent methods using deep learning have shown some success in approximating optimal auctions, recovering several known solutions and outperforming strong baselines when optimal auctions are not known. In addition to maximizing revenue, auction mechanisms may also seek to encourage socially desirable constraints such as allocation fairness or diversity. However, these philosophical notions neither have standardization nor do they have widely accepted formal definitions. In this paper, we propose PreferenceNet, an extension of existing neural-network-based auction mechanisms to encode constraints using (potentially human-provided) exemplars of desirable allocations. In addition, we introduce a new metric to evaluate an auction allocations' adherence to such socially desirable constraints and demonstrate that our proposed method is competitive with current state-of-the-art neural-network based auction designs. We validate our approach through human subject research and show that we are able to effectively capture real human preferences. Our code is available on GitHub.

## 1 Introduction

Auctions are an essential tool in many marketplaces, including those in electricity [8], advertising [14], and telecommunications [9, 26]. The design of auctions with desirable properties is thus an important theoretical and practical problem. A typical assumption is that bidders may choose to bid strategically and will successfully anticipate the behavior of other bidders. This results in a potentially complicated Bayes-Nash equilibrium which may be difficult to predict. To evade this problem, a common requirement is that an auction should be strategyproof (i.e. bidders should be incentivized to truthfully share their valuations regardless of other bidders' actions).

If the goal of an auction is to maximize the total welfare of all participants, the Vickrey-Clarke-Groves (VCG) mechanism is both strategyproof and welfare-maximizing [40, 7, 20]. Intuitively, in many cases the VCG mechanism corresponds to a second-price auction. If the goal is to maximize the revenue gained, the problem is more challenging. Myerson's work completely characterizes strategyproof, revenue-maximizing auctions for a single item [29]. Beyond this case, results are more limited. Some results are known for the "multiple-good monopolist" problem, in which the auctioneer sells multiple items to a single bidder [11, 12, 24, 31, 28, 21, 18]. In addition, designing auctions for the related but weaker notion of Bayes-Nash incentive compatibility is reasonably well understood [2, 3, 4].

There has been significant difficulty in designing strategyproof auctions involving multiple items and multiple bidders. [41] shares significant, but limited results which solve the case in which items

---

\* The first two authors contributed equally to this work. 35th Conference on Neural Information Processing Systems (NeurIPS 2021).

may have at most 2 values. Due to the apparent difficulty of analytically designing strategyproof, revenue maximizing auctions, recent methods instead approximate optimal auctions using machine learning approaches [13, 10, 19, 25, 15, 33, 32]. [37] proposes an method which guarantees exact strategyproofness in the single-agent setting. [39, 1] also use neural networks in the design of auctions. These methods primarily focus on standard mechanism design goals of revenue or welfare maximization, but these may not be the only goals of an auction. An auction might be conducted for public goods (i.e. electromagnetic spectrum distribution [9]), where the auctioneer must consider the effect not only on the auctioneer and bidders but on third-parties as well [30]. In other cases, such as auctions involving job or credit advertisements, it might be necessary to additionally constrain the auction mechanism to ensure fairness with respect to protected characteristics [23, 6]. Recent work has considered the problem of determining revenue-maximizing, strategyproof, fair auctions, either from a specific class [5] or with a general neural network approach [25].

The underlying notions of "fairness" used in these papers (e.g. total-variation fairness [23]) are defensible but somewhat arbitrary – they are attempts to formally capture human intuition. It might instead be better to attempt to capture, from data, human intuitions about fairness or other ethical requirements that are not explicitly defined [38, 16].

In this paper, we introduce PreferenceNet, an extension to RegretNet that directly encodes socially desirable constraints from data, and captures noisy human preferences. We are motivated by advertising auctions where the allocations of the auction mechanism must satisfy human preference constraints alongside the typical goals of strategyproofness and revenue maximization. We conduct a number of experiments using synthetic data (as is typical for neural network based auction mechanisms [13]) and evaluate our method on different auction settings and fairness constraints. We show that PreferenceNet is able to effectively capture each fairness constraint and matches the performance of standard approaches. We also conduct two surveys to further study human preferences. In the first survey (n=140), when given a specific definition of fairness, we ask participants to determine if a given auction setting is fair in order to test the noise in human judgements. In the second survey (n=345), we elicit judgements of pairwise comparisons of two auction settings to determine preferences without priming the participants with any particular definition of fairness. We train PreferenceNet on these data and show that our approach can capture nuanced human preferences in auction design.

## 2 Background and Related Work

**RegretNet.** [13] presents RegretNet, a neural-network architecture for learning approximately strategyproof auctions that maximize revenue. RegretNet treats the auction mechanism as a function from bids to allocations and payments, parameterized as a neural network. Revenue is optimized via gradient descent on sampled truthful bid profiles; strategyproofness is enforced by computing strategic bids in an adversarial manner to minimize violations. RegretNet has been modified and extended in a variety of ways, particularly to enforce additional desirable constraints. [10, 39, 19, 25, 15, 33].

**Special Case: Single-Bidder Auctions.** We highlight a special case in which deep learning for auctions has been particularly successful – auctions with a single bidder. In the single-bidder setting, the set of strategyproof mechanisms can be easily characterized [34], and it is possible to design neural network architectures which will always lie in this set. [13, 37] present two different learning-based solutions. Both methods are guaranteed to be strategyproof, and revenue can be maximized by unconstrained optimization. There are some known optimal single-agent mechanisms (we highlight those of Manelli-Vincent [28] and Pavlov [31], as they are relevant to auction settings we study below). Moreover, the theory of single-bidder mechanisms is well understood [11, 12, 18], so it is also possible to learn empirically strong mechanisms using these approaches and then prove their optimality [37, 13]. Unfortunately, when moving beyond the single-agent setting, it is necessary to use more general neural network architectures for which these guarantees do not apply. Because we are interested in such settings, we focus on these general architectures in this work.

**Fairness and Human Preferences.** As mentioned, while revenue, strategyproofness, and individual rationality are the classic goals of auction design, it might also be necessary for allocations made by an auction to satisfy certain other requirements such as fairness [5, 25]. However, it is not always

clear if these mathematical definitions of these concepts actually capture human intuitions. [36, 38] considers human reactions to different definitions of fairness in a classification setting. [35] tests the extent to which human participants are able to understand and apply fairness metrics. [16] considers the problem of learning to perform fair clustering from human-provided demonstrations. As discussed below, we also crowdsource human opinions on fairness in auctions and analyze the results in Section 6.

# 3   Problem Setting

**Auction Model.**   An auction is defined as a set of agents $N = \{1, \ldots, n\}$ bidding for items $M = \{1, \ldots, m\}$. Each agent $i \in N$ has a corresponding private valuation $v_i$, randomly drawn from a set of $n$ valuations as $v = (v_1, \ldots v_n) \in V_i$. In general $v_i$ may be functions over the power set of items $2^M$. However, we only consider simpler cases with **additive** valuations and **unit-demand** valuations, where the valuation is simply a vector $v_i \in \mathbb{R}^m$ of values per item.

Each agent reports a bid vector $b_i$ to the auctioneer, which may differ from the private valuation $v_i$. Given the profile of bids $b = (b_1, \ldots, b_n)$ of all agents, the auction has allocation and payment rules $g(b) : \mathbb{R}^{mn} \to [0, 1]^{nm}$ and $p(b) : \mathbb{R}^{mn} \to \mathbb{R}^n$. We will refer to the matrix of allocation probabilities, whose rows must sum to 1, as $g(b) = z$ . Likewise agent $i$'s value of the $j$th item is $v_{i,j}$, and bidder $i$ has payment function $p_i$. Moreover, for unit-demand auctions, we restrict the allocation to allow each bidder to win, in expectation, at most 1 item. Given the allocation, each agent receives a utility which can in either case be represented in linear form as $u_i(v) = \sum_j v_{i,j} z_{i,j} - p_i$.

**Desirable Auction Properties.**   A mechanism is individually rational (IR) when an agent is guaranteed non-negative utility: $u_i(v_i; v) \geq 0 \ \forall i \in N, v \in V$. A mechanism is dominant-strategy incentive-compatible (DSIC) or strategyproof if every agent maximizes their own utility by bidding truthfully, regardless of the other agents' bids. We can define regret, the difference in utility between the bid player $i$ actually made and the best possible strategic bid: $\mathrm{rgt}_i(v) = \max_{b_i} u_i(b_i, v_{-i}) - u_i(v_i, v_{-i})$. In addition to satisfying the IR and DSIC constraints, the auctioneer seeks to maximize their expected revenue. If the auction is truly DSIC, players will bid truthfully, and as a result revenue is simply $E_{v \sim V}[\sum_{i \in N} p_i(v)]$.

# 4   PreferenceNet: Encoding Human Preferences

We first explore a new metric to evaluate the adherence to socially desirable constraints in item allocations. Next, we describe the implementation details of PreferenceNet and important considerations when training the model.

**Evaluation Metrics.**   There are limited evaluation criteria that quantitatively measures an auction mechanism's ability to enforce constraints on item allocations. If the constraints are not known explicitly, one can qualitatively examine the allocation graphs to evaluate the underlying allocation function $g(b)$. However, visual inspection does not scale to larger auction settings. As shown in Figure 1, our proposed metric is not only able to capture the same insights as qualitative analysis, but also scales to arbitrarily large auction mechanisms.

Given the limitations of existing analysis techniques, we propose Preference Classification Accuracy (PCA), a new metric to evaluate how well a learned auction model satisfies an arbitrary constraint. For a set of test bids $b$ and allocations $g(b) : \mathbb{R}^{mn} \to [0, 1]^{nm}$ generated by our learned auction model, we assign a label $s(b) \in \{1, 0\}$ to each allocation according to a ground truth labeling function based on the underlying preference. For each test bid $b$, $s(b)$ is 1 if the learned auction network satisfies the ground truth constraint. PCA is calculated by averaging the value of $s(b)$ over $n$ test bids. Note that this metric remains valid in cases when we know the underlying preference function (e.g. total variation fairness, entropy) as well as when we are sampling from an unknown distribution (e.g. human preference elicitation). For cases where the underlying preference function is known, we can directly compute the preference score for a given allocation and apply a threshold to obtain a label. For cases where the preference function is unknown, as is the case in human preference elicitation, we can use the ground truth allocation-label pair to assign preference labels $s(b)$ to new allocations based on the nearest neighbor in the ground truth set. This metric gives us a formal way to measure

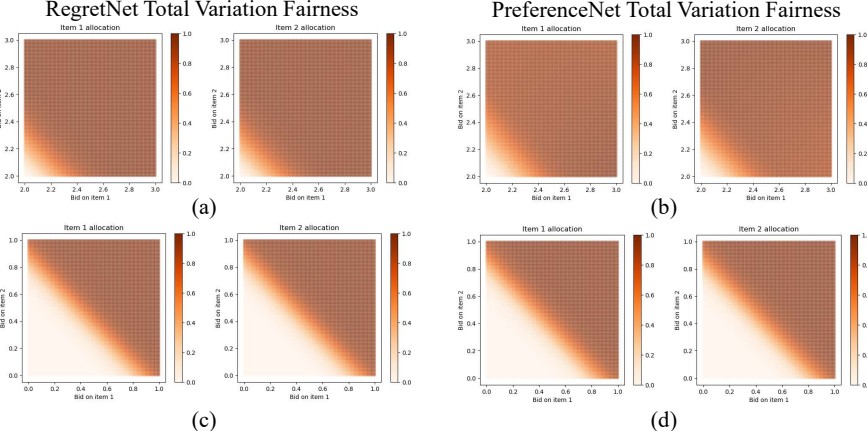

Figure 1: We compare the allocation plots of standard RegretNet approaches in enforcing total variation fairness (TVF) [25, 23] (sub-figures a and c) with our proposed approach (sub-figures b and d) learned through exemplars of desirable allocations that satisfy TVF. Visually, the allocations for both the unit-demand (sub-figures a and b) and additive auctions (sub-figures c and d) are identical. In this case, our proposed metric verifies our visual inspection, indicating that allocations from all four models satisfy TVF with 100% accuracy. However, this qualitative analysis does not extend to larger auction settings. In contrast, our proposed metric allows us to quantify the adherence of an auction mechanism to an enforced constraint for arbitrarily large auctions.

the degree to which preference constraints are violated, which crucially can be used whether or not the constraints follow an explicitly-known function.

**Preference Elicitation.** In order to effectively elicit preferences, we rely on pairwise comparisons between allocations to identify both positive and negative exemplars. We compare each input set of of allocations against $n$ other allocations to determine if a particular sample is preferred over these alternatives (see Figure 2). This method of group preference elicitation reduces noise and ensures that the learned preference is satisfactory to a majority of the participants. We use this ranking approach to generate training labels in all of our experiments as described below.

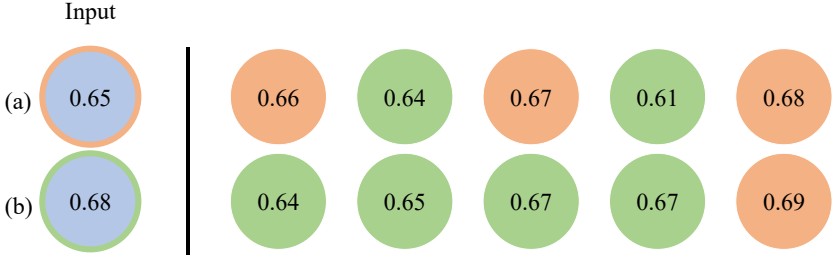

Figure 2: To elicit preferences from a group, or from a person without deterministic preferences, we use pairwise comparisons to determine labels for each allocation. In examples (a) and (b), each allocation (represented as a circle) has a preference score (represented by a number inside the circle). We can compare the score of the input against $n$ other valuations to determine the relative ranking of the input data point. If the input data point has a smaller score than a plurality of the points it is compared against, it is a negative exemplar for the implicit preference (as in (a)). Otherwise it is positive (as in (b)).

**Training Algorithm.** We present the training algorithm of PreferenceNet below. We use the same additive and unit-demand network architectures as RegretNet for arbitrary numbers of agents and items. Our training algorithm follows closely from RegretNet. PreferenceNet consists of two sub-networks: RegretNet and a 3-layer MLP with a sigmoid activation at the output. We first train the MLP using a uniformly drawn sample of allocations as inputs, and optimize the binary cross entropy loss function using ground truth labels (generated as in Figure 2) identifying positive and negative exemplars. Next, we train RegretNet using the standard training procedure with the modified loss function described in Eq. 1. Lastly, we sample allocations and payments from the partially trained RegretNet model every $c$ epoch and augment the MLP training set to adapt to the distributional shifts in allocations over the course of training. Our modified loss function $\mathcal{C}_\rho(w; \lambda)$ is defined as:

$$\mathcal{L}_{\mathrm{rgt}} = \sum_{i \in N} \lambda_{(r,i)} \, \mathrm{rgt}_i(w) + \frac{\rho_r}{2} \sum_{i \in N} \mathrm{rgt}_i(w)^2, \; L_{\mathrm{s}} = \sum_{j \in M} \mathrm{s}_j$$

$$\mathcal{C}_\rho(w; \lambda) = -\frac{1}{L} \sum_{l=1}^{L} \sum_{i \in N} p_i^w(v^{(l)}) + \mathcal{L}_{\mathrm{rgt}} - \mathcal{L}_{\mathrm{s}} \tag{1}$$

where $\mathcal{L}_{\mathrm{s}}$ is the output of the trained MLP. RegretNet is optimized such that it is strategyproof, revenue-maximizing, satisfies the preference learned by the MLP (i.e. maximizes the output scores of the MLP).

For each configuration of $n$ agents and $m$ items, we train RegretNet for a maximum of 200 epoch using 160,000 training samples. We also train the MLP with 80,000 initial training samples and iteratively retrain the MLP with 5,000 additional samples from the partially trained RegretNet every 5 epoch. For all networks, we use the Adam optimizer and 100 hidden nodes per layer. We apply warm restarts to the MLP optimizer each time we add new training data to to prevent the model from settling in a local minima. We incremented $\rho_r$ every 2500 iterations and $\lambda_r$ every 25 iterations. Finally, we report the preference classification accuracy, mean regret, and mean payments by simulating the allocations of 20,000 testing samples. We run all our experiments on an NVIDIA Titan X (Pascal) GPU. We refer readers to GitHub for our implementation.

**MLP Architecture.** We learn implicit preference functions using a simple 3-layer multi-layer perceptron that takes as input a mini-batch of allocations and outputs a vector that scores the input $\in [0,1]$ as a measure of how closely the allocation satisfies the ground truth preference. Given that neural networks are universal function approximators, with enough parameters the MLP can represent any arbitrary preference function. In practice, we find that ReLU non-linear activations and batch normalization are essential to effectively train this network.

**Class Balanced Sampling.** The distribution of positive and negative training examples for arbitrary preferences are often imbalanced – preferred examples may be concentrated in a small region of possible allocations. Often, this imbalance can make it difficult to train a robust model. Commonly, neural networks with improper class balance fail to learn a good decision boundary. In order to compensate for such imbalanced datasets, we can explicitly over-sample sparse classes until there are an equal number of positive and negative training examples. In practice this allows the network to quickly learn the decision boundary and allows us to train with fewer data points.

**MLP Co-Training.** The distribution of payments and allocations generated by RegretNet shift significantly during the course of training. As a result, the initially trained MLP may not effectively enforce the preference loss as RegretNet continues to train. In order to adapt to the distribution shift, we sample allocations from RegretNet at fixed intervals while it is training, add them back to the training set, and retrain the MLP. We generate noisy labels [27] using the existing MLP to reduce the cost of collecting expensive labels. In practice, we find that this approach effectively reinforces the decision boundary between positive and negative exemplars.

**Model Validation and Selection.** The optimal model minimizes regret, while maximizing payments and preference classification accuracy. In practice, satisfying all three conditions may be difficult. Often, we find that choosing a fixed epoch to evaluate does not provide consistent results. Rather, we evaluate each checkpoint on a validation set and maximize the following criteria in Eq. 2:

$$\alpha * \mathrm{PCA} + \beta * \frac{\bar{p}(b)}{\max(p(b))} + \gamma * (1 - \frac{\bar{\mathrm{rgt}}(b)}{\max(\mathrm{rgt}(b))}) \text{ s.t. } \alpha + \beta + \gamma = 1 \tag{2}$$

In our experiments we set $\alpha = 0.45, \beta = 0.1, \gamma = 0.45$. It is important to note that the maximum regret and payments are calculated over the entire training process, while the mean regret, payment, and preference classification accuracy are calculated at each epoch.

# 5  Sampling Synthetic Preferences

Given the lack of publicly available auction data, we generate synthetic bids as in [13, 25, 37]. We first validate PreferenceNet using synthetic preferences, and extend our analysis to real human preferences in Section 6. In our synthetic preference experiments, we are interested in three types of fairness: total variation fairness (TVF), entropy, and a quota system. All three definitions of fairness map the allocations $g(b)$ onto $\mathbb{R}$. We train auction models for each of these valuation function and compare RegretNet with our proposed model under uniform additive and unit-demand auction settings in Table 5. Note in Table 5 that RegretNet is trained with an additional loss function term to explicitly optimize for the preference. PreferenceNet is trained as described in 4. We describe the three definitions of fairness below:

**Total Variation Fairness.** An auction mechanism satisfies total variation fairness if the $\ell_1$-distance between allocations for any two users is at most the distance between those users. That is, total variation fairness is satisfied when

$$\forall k \in \{1, ..., c\}, \forall j, j' \in M, \sum_{i \in C_k} |g(b)_{i,j} - g(b)_{i,j'}| \le d^k(j, j').  \qquad (3)$$

We fix $d = 0$ in all of our experiments. We minimize violations of the above constraints.

**Entropy.** An auction mechanism satisfies the entropy constraint if the entropy of the normalized allocation for a bid profile $b$

$$-\sum_{i=1}^{n} P\left(\frac{g(b)_{i,\cdot}}{\sum_j g(b)_{ij}}\right) \log P\left(\frac{g(b)_{i,\cdot}}{\sum_j g(b)_{ij}}\right)  \qquad (4)$$

is maximized. We normalize across items, turning the allocation into a probability distribution. This normalization ensures that entropy reflects diversity in the allocations, and not the overall number of items being allocated. Specifically, encouraging entropy ensures that the allocation for a given agent will tend to be more uniformly distributed.

**Quota.** An auction mechanism satisfies the quota constraint if, for each item in the (normalized) allocation, the smallest allocation to any agent $j$ is greater than some minimum threshold $t$:

$$\min_j \left(\frac{g(b)_{\cdot,j}}{\sum_i g(b)_{i,j}}\right) > t  \qquad (5)$$

Here, we normalize the allocation so that the allocation per item will be a probability distribution over agents. The intuition for this definition is that each agent must guarantee some floor across every item. Returning to the advertising example, this ensures some minimum percentage of ad impressions are seen by every demographic group.

We train a number of models to compare the performance between RegretNet and PreferenceNet. Despite leveraging an weaker, implicit signal to learn fairness constraints, PreferenceNet is able to closely match the performance of RegretNet. Surprisingly, PreferenceNet improves upon RegretNet in some cases, indicating that with careful hyperparameter tuning, further improvements are possible. Of the three fairness constraints examined, enforcing a quota is hardest for additive auctions, as both RegretNet and PreferenceNet struggled for auctions with a large number of agents.

**Limitations.**  Despite its effectiveness, we highlight limitations of our approach. In general, PreferenceNet always optimizes for the simplest function. For example, if learning a piece-wise preference where the positive exemplars are not clustered in a single region as in Figure 3, PreferenceNet tends to only satisfy part of the piece-wise function. This is unsurprising, given that neural networks are known to take shortcuts in optimization [17]. Moreover, PreferenceNet has difficulty in generating allocations with tightly clustered preference scores to satisfy a particular constraint. Given that we optimize for preferences implicitly using exemplars, this behavior is understandable. We study these limitations further in the supplemental material.

Table 1: We evaluate both RegretNet and PreferenceNet over $n$x$m$ auctions ("u" refers to unit-demand and "a" refers to additive), where $n$ is the number of agents and $m$ is the number of items. We measure three criteria **(a) PCA**, **(b) Regret Mean (STD)**, **(c) Payment Mean (STD)**. Although PreferenceNet learns each preference implicitly, it produces similar performance to our strong baseline.

| (a) | TVF | | Entropy | | Quota | |
|---|---|---|---|---|---|---|
| | RegretNet | PreferenceNet | RegretNet | PreferenceNet | RegretNet | PreferenceNet |
| 2x2 u | 100.0 | 100.0 | 86.4 | 69.6 | 100.0 | 100.0 |
| 2x4 u | 100.0 | 100.0 | 99.7 | 94.9 | 100.0 | 100.0 |
| 4x2 u | 100.0 | 100.0 | 100.0 | 94.5 | .1 | 100.0 |
| 4x4 u | 100.0 | 99.3 | 100.0 | 67.4 | 100.0 | 100.0 |
| 2x2 a | 100.0 | 100.0 | 99.6 | 99.5 | 75.1 | 100.0 |
| 2x4 a | 100.0 | 100.0 | 100.0 | 100.0 | 36.0 | 100.0 |
| 4x2 a | 100.0 | 100.0 | 99.9 | 100.0 | .1 | .1 |
| 4x4 a | 100.0 | 99.9 | 100.0 | 96.1 | 0 | 0 |

| (b) | TVF | | Entropy | | Quota | |
|---|---|---|---|---|---|---|
| | RegretNet | PreferenceNet | RegretNet | PreferenceNet | RegretNet | PreferenceNet |
| 2x2 u | .012 (.012) | .012 (.012) | .02 (.02) | .011 (.01) | .012 (.013) | .013 (.012) |
| 2x4 u | .008 (.006) | .016 (.013) | .045 (.045) | .022 (.019) | .012 (.01) | .034 (.022) |
| 4x2 u | .015 (.009) | .028 (.013) | .025 (.025) | .056 (.018) | .026 (.017) | .819 (.136) |
| 4x4 u | .033 (.024) | .067 (.037) | .031 (.031) | .029 (.014) | .037 (.023) | .432 (.145) |
| 2x2 a | .005 (.004) | .006 (.004) | .006 (.006) | .013 (.008) | .008 (.007) | .05 (.031) |
| 2x4 a | .008 (.011) | .008 (.009) | .007 (.007) | .008 (.009) | .01 (.01) | .078 (.032) |
| 4x2 a | .038 (.076) | .014 (.011) | .036 (.036) | .162 (.052) | .017 (.013) | .017 (.013) |
| 4x4 a | .424 (.324) | .139 (.104) | .015 (.015) | .257 (.1) | .038 (.017) | .039 (.018) |

| (c) | TVF | | Entropy | | Quota | |
|---|---|---|---|---|---|---|
| | RegretNet | PreferenceNet | RegretNet | PreferenceNet | RegretNet | PreferenceNet |
| 2x2 u | 4.18 (.45) | 4.17 (.44) | 4.26 (.37) | 4.14 (.43) | 4.19 (.35) | 4.18 (.33) |
| 2x4 u | 4.37 (.29) | 4.41 (.34) | 4.48 (.34) | 4.47 (.21) | 4.44 (.16) | 4.49 (.29) |
| 4x2 u | 4.93 (.21) | 4.98 (.21) | 4.85 (.23) | 4.99 (.21) | 5.16 (.24) | 5.03 (.21) |
| 4x4 u | 8.81 (.39) | 8.92 (.38) | 8.79 (.5) | 8.81 (.42) | 8.8 (.3) | 8.81 (.36) |
| 2x2 a | .87 (.31) | .87 (.32) | .88 (.32) | .9 (.31) | .73 (.37) | .6 (.3) |
| 2x4 a | 1.75 (.38) | 1.74 (.4) | 1.77 (.45) | 1.74 (.4) | 1.76 (.44) | 1.39 (.5) |
| 4x2 a | 1.1 (.34) | 1.2 (.22) | 1.1 (.34) | 1.15 (.22) | 1.3 (.23) | 1.31 (.23) |
| 4x4 a | 2.58 (.39) | 2.41 (.36) | 2.26 (.3) | 2.28 (.32) | 2.52 (.32) | 2.56 (.33) |

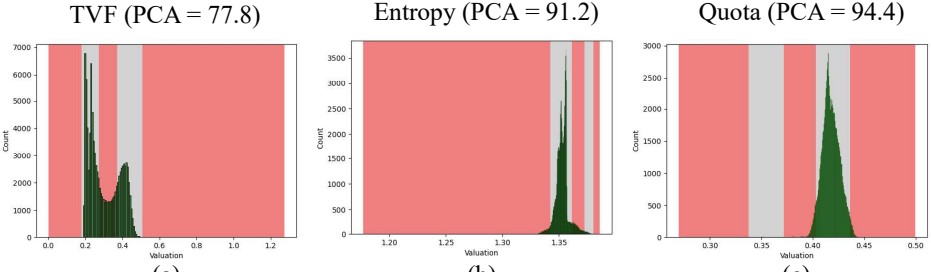

Figure 3: We simulate preference elicitation where positive exemplars are spread along multiple bands. The grey bands represent ground truth regions of positive exemplars, and the red bands represent ground truth regions of negative exemplars. In each plot, the green histogram represents the preference scores of the generated allocations. A model that perfectly enforcing a given preference rule will generate allocations that have valuations entirely within the grey bands.

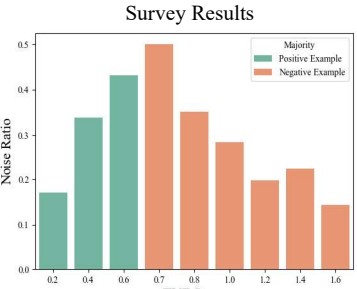
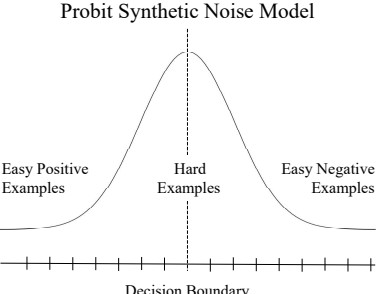

Figure 4: We crowdsource human annotators to examine various advertising scenarios and determine if an allocation is fair according to a given definition. We simplify the definition of total variation fairness (TVF) and measure the noise in responses as a function of the ambiguity of the scenario. We expect that the the label noise for a particular TVF value is inversely proportional to the distance from the decision boundary. Concretely, easy allocations will have lower noise ratios, and ambiguous allocations will have high noise ratios. We can leverage this model of uncertainty and apply it to our synthetic experiments to better simulate human preference elicitation.

## 6 Soliciting Human Preferences

In this section, we detail the creation of and results derived from two human subjects surveys. The results of these surveys are used to validate core assumptions of our model and provide data that we use to train an auction model. We find that PreferenceNet is able to adhere to the notion of fairness expressed by the human's preferences of auction outcomes. This encouraging result validates the utility and expressiveness of PreferenceNet.

We conducted both surveys through Cint, a crowdsourcing platform which connected us with English-speaking participants located in the United States. After submitting an application for our human subjects research to our institutional IRB, we were notified that the survey was exempt from IRB review. Our survey protocols can be found in the supplemental materials. Cint compensates per survey completion (regardless of length to complete), and both surveys were set to pay $1.92. All survey results are anonymized to protect participant privacy. We include these results in the supplemental material.

**Measuring Preference Noise.** Our first survey was designed to test how participants would interpret and apply a simple fairness definition. The survey protocol was designed as follows: We primed participants to consider an advertising auction that was shown to two different (vague) demographic groups. Each participant was told that an auction would be fair if each ad was presented to each group at equal rates. After some familiarization, we asked the participants to determine if a given scenario was fair. Each participant was asked 30 such questions. The median completion time for the survey was 6 minutes with a median hourly wage of $18. The 30 questions presented to the participants came from a question bank of 64 randomly generated scenarios, each with an associated TVF score.

Human understanding is an inherently noisy process. Despite providing the same context, we observe that survey participants understand a given definition of fairness and apply it to various scenarios differently. Given this data, we perform a normality test using a Q-Q plot as shown in the supplemental material. We find that our survey data are well correlated with the Gaussian distribution. This is also well supported by our visual inspection of the noise distribution. Experimentally, we observe that participants' choice of what is fair has a decision boundary at approximately a TVF value of 0.7. Interestingly, the noise is maximal near the decision boundary as shown in Figure 4. Concretely, we can model this noise using a probit model, so that the probability that that the label is unpertubed for a particular TVF value increases with distance such that $P(Y = 1|X) = k\Phi(\frac{|x-\mu|}{\sigma})$, where $\Phi$ is the CDF of the Gaussian distribution, $\mu$ is the decision boundary, $\sigma$ is the measured sample standard deviation, and $k$ is an optional parameter that can scale the noise. We can use this noise model to perturb the input training data to the MLP to better simulate real data. We explore this further in the supplemental material.

Unit Demand Auction

| | Human | TVF | Entropy | Quota |
|---|---|---|---|---|
| Human | 0 | 0.005 | 0.004 | 0.009 |
| TVF | 0.005 | 0 | 0.002 | 0.011 |
| Entropy | 0.004 | 0.002 | 0 | 0.010 |
| Quota | 0.009 | 0.011 | 0.010 | 0 |

Additive Auction

| | Human | TVF | Entropy | Quota |
|---|---|---|---|---|
| Human | 0 | 0.002 | 0.002 | 0.032 |
| TVF | 0.002 | 0 | 0.001 | 0.033 |
| Entropy | 0.002 | 0.001 | 0 | 0.032 |
| Quota | 0.032 | 0.033 | 0.032 | 0 |

| Human Preferences | PCA | Regret Mean (STD) | Payment Mean (STD) |
|---|---|---|---|
| 2x2 Unit | 100.0 | .016 (.015) | 4.20 (.37) |
| 2x2 Additive | 100.0 | .005 (.004) | .87 (.31) |

Figure 5: We randomly sample 20,000 bids and compute the average L2 distance from each model's learned allocations to identify allocation similarity. We find that human preferences are most similar to both TVF and entropy in both the unit demand and additive auction settings. Moreover, PreferenceNet is able to perfectly capture human preferences with low regret.

**Group Preference Elicitation.** We designed our second survey to ask similar questions to the preference noise survey above. However, there were two primary differences: (1) we did not prime the participants with a fairness definition, and (2) the participants were presented with two scenarios and were asked which they thought was more fair. Each participant was asked 30 of these pairwise questions on the questions described above (full protocol details in the supplemental materials). This survey had 345 participants who had a median completion time of 7 minutes and a median pay of $15 per hour.

Notions of fairness and diversity have neither standardized nor widely accepted formal definitions [38, 35, 36, 22]. The purpose of this survey is to elicit preferences from a group and train an auction model whose allocations resemble the group preference. Using the survey data, we apply our preference elicitation strategy as described in Section 4 to generate training labels for each sample. After training PreferenceNet to enforce the group preference for both the unit-demand and additive auction settings, we find that the group preference is more similar to TVF and entropy. As shown in Figure 5, human preferences are not perfectly captured by these typical models, both because group preferences rarely converge to a unifying model, and preference elicitation is a noisy process.

# 7 Conclusion

Although surveying people to elicit their preferences can effectively help us model ambiguous definitions of socially beneficial constraints, we must be careful about the framing of the survey questions and the choice of audiences we survey.

**Ethical Implications.** Humans are inherently biased, so we need to be cognizant of the effects these latent biases might have over preferences for fairness. Moreover, sampling human preferences facilitates opportunities for data poisoning attacks, in which a malicious survey respondent could try to negatively impact the survey collection process. In general, we can mitigate both of these issues by sampling at scale to avoid problems with noisy labels, although this brings additional cost. Most importantly, we must involve stakeholders to ensure that their preferences are validated through the learned model in an iterative process.

In this paper we present PreferenceNet, a novel extension to RegretNet that makes it easier to learn preferences from data to encode socially desirable constraints for auction design. We introduce a new metric to empirically measure how closely a learned mechanism enforces a particular constraint, and show that our proposed method is able to effectively capture human preferences.

## Acknowledgments and Disclosure of Funding

This research was supported in part by NSF CAREER Award IIS-1846237, NSF D-ISN Award #2039862, NSF Award CCF-1852352, NIH R01 Award NLM-013039-01, NIST MSE Award #20126334, DARPA GARD #HR00112020007, DoD WHS Award #HQ003420F0035, and a Google Faculty Research Award. We thank the authors of ProportionNet for sharing their codebase and Kevin Kuo and Uro Lyi for their feedback in writing this paper.

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
