# A Simulating Noisy Preferences

We revisit survey results from Section 6, and explore the impact of label noise on our proposed method. We show that despite significant input perturbation, PreferenceNet is able to effectively capture preferences from noisy labeled examples.

**Comparing Survey Data to Probit Model.** In order to study noise in preference elicitation, we ask survey participants to interpret and apply a simple fairness definition. We primed participants to consider an advertising auction that was shown to two different (vague) demographic groups. Each participant was told that an auction would be fair if each ad was presented to each group at equal rates. After some familiarization, we asked the participants to determine if a given scenario was fair.

We hypothesize that the distribution of noise should be distributed according to a probit model, defined as $k\Phi\left(\frac{|x-\mu|}{\sigma}\right)$, where $\Phi$ is the CDF of the Gaussian distribution, $\mu$ is the decision boundary, $\sigma$ is the measured sample standard deviation, and $k$ is an optional parameter that can scale the noise. The intuition behind this particular noise model is that participants will have higher uncertainty about allocation fairness closer to the decision boundary, and lower uncertainty farther away from the decision boundary. In practice, the probit model closely follows the survey noise distribution close to the sample mean, but diverges farther away from the decision boundary. The assumption that label noise goes to zero at sufficient distance from the mean does not hold in our human subject research. Rather, we find that there is a minimum amount of uncertainty regardless of the distance from the decision boundary, indicating that human understanding of fairness is inherently noisy.

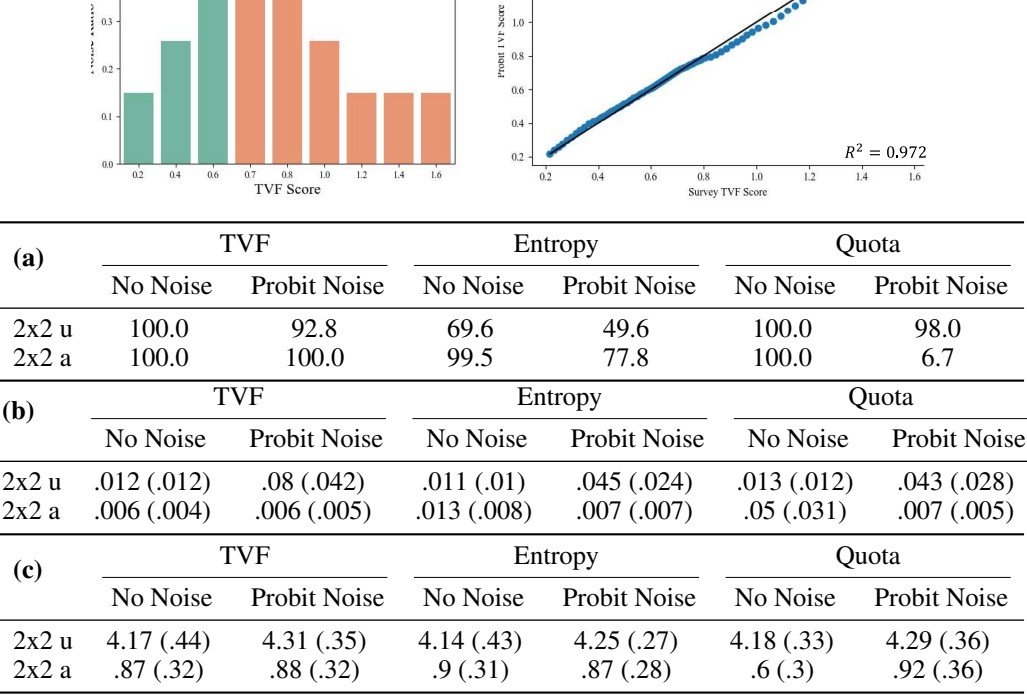

| **(a)** | TVF | | Entropy | | Quota | |
|---|---|---|---|---|---|---|
| | No Noise | Probit Noise | No Noise | Probit Noise | No Noise | Probit Noise |
| 2x2 u | 100.0 | 92.8 | 69.6 | 49.6 | 100.0 | 98.0 |
| 2x2 a | 100.0 | 100.0 | 99.5 | 77.8 | 100.0 | 6.7 |

| **(b)** | TVF | | Entropy | | Quota | |
|---|---|---|---|---|---|---|
| | No Noise | Probit Noise | No Noise | Probit Noise | No Noise | Probit Noise |
| 2x2 u | .012 (.012) | .08 (.042) | .011 (.01) | .045 (.024) | .013 (.012) | .043 (.028) |
| 2x2 a | .006 (.004) | .006 (.005) | .013 (.008) | .007 (.007) | .05 (.031) | .007 (.005) |

| **(c)** | TVF | | Entropy | | Quota | |
|---|---|---|---|---|---|---|
| | No Noise | Probit Noise | No Noise | Probit Noise | No Noise | Probit Noise |
| 2x2 u | 4.17 (.44) | 4.31 (.35) | 4.14 (.43) | 4.25 (.27) | 4.18 (.33) | 4.29 (.36) |
| 2x2 a | .87 (.32) | .88 (.32) | .9 (.31) | .87 (.28) | .6 (.3) | .92 (.36) |

Figure 6: We measure the performance of PreferenceNet with label noise that follows the Probit model for 2 agent, 2 item auctions ("u" refers to unit-demand and "a" refers to additive). We measure three criteria **(a) PCA**, **(b) Regret Mean (STD)**, **(c) Payment Mean (STD)**. We find that PreferenceNet is robust to noise in most cases, despite $25\%$ of labels being flipped. PCA is most sensitive to noise. We find that the level of performance degradation is dependent on the particular constraint and auction type. In contrast, average regret and payments are comparable irrespective of input label noise.

We can update our probit model to incorporate this fact. We instead estimate the probability of a label flip as $min(k\Phi(\frac{|x-\mu|}{\sigma}), f)$, where $f$ represents the noise floor. We experimentally select $k = 1.05$ and $f = 0.15$ to minimize the difference between the real and synthetic distributions. We compare the real distribution to the probit model in Figure 6 using a Q-Q plot. Since we expect the synthetic noise model to closely model the survey noise distribution, we plot the quantile function in comparison to the line $y = x$ (in black), and find that our synthetic model closely models real noise ($R^2 = 0.972$).

We train PreferenceNet, perturbing labels according to their distance from the decision boundary. Despite perturbing more than $25\%$ of the training labels, PreferenceNet is able to capture the underlying preference, while minimizing regret and maximizing revenue.

## B   Mixing Preferences in Synthetic Experiments

Eliciting preferences from a group is particularly challenging, because these preferences often heterogeneous. In Section 6 we train PreferenceNet on real human preferences and find that we can capture human preferences with high accuracy. To further study group preference elicitation, we train several models on a mixture of synthetic preferences to better understand the results of our human subject experiments. Specifically, the training set for the MLP contains allocations drawn from a uniform distribution, which are partitioned into 3 sets and labeled according to a particular definition.

As mentioned in Section 5, PreferenceNet often optimizes for the simplest function. In Table 2, we can see that almost all models maximize PCA for Entropy and TVF over a quota system irrespective of the mixture of training labels. In general, there does not seem to be a clear correlation between the input weighting of the different preferences and the learned preference function. This may explain why PreferenceNet trained on human preferences generates allocations that optimize for TVF and entropy.

Table 2: We simulate preference elicitation of three different definitions of fairness. For a set of allocations, we label non-overlapping partitions according to different preference functions. We calculate the PCA of each constraint independently, and weight them according to the proportions of the training labels. For unit demand auctions, we find that all three constraints have high PCA, indicating that PreferenceNet is able to generate a set of allocations that satisfy all preferences. However, we find that additive auctions have inconsistent results, indicating the ability to encode group preferences in auction allocation is dependent on the auction type.

| Mixture | TVF PCA | Entropy PCA | Quota PCA | Average PCA |
|---|---|---|---|---|
| 50% TVF, 25% Entropy, 25% Quota u | 99.7 | 100.0 | 99.4 | 99.9 |
| 25% TVF, 50% Entropy, 25% Quota u | 93.9 | 100.0 | 93.6 | 98.6 |
| 25% TVF, 25% Entropy, 50% Quota u | 100.0 | 100.0 | 100.0 | 99.6 |
| 33.3% TVF, 33.3% Entropy, 33.3% Quota u | 98.3 | 100.0 | 98.3 | 99.4 |
| 50% TVF, 25% Entropy, 25% Quota a | 61.7 | 87.5 | 94.2 | 78.8 |
| 25% TVF, 50% Entropy, 25% Quota a | 99.4 | 100.0 | 99.9 | 99.9 |
| 25% TVF, 25% Entropy, 50% Quota a | 15.0 | 15.3 | 18.0 | 16.0 |
| 33.3% TVF, 33.3% Entropy, 33.3% Quota a | 59.4 | 75.7 | 57.5 | 64.4 |

## C   Augmented Lagrangian Multipliers to Enforce Constraints

In this section we explore the impact of enforcing constraints explicitly using augmented lagrangian multipliers. We we modify Eq. 1 as follows:

$$\mathcal{L}_{\text{rgt}} = \sum_{i \in N} \lambda_{(r,i)} \, \text{rgt}_i(w) + \frac{\rho_r}{2} \sum_{i \in N} \text{rgt}_i(w)^2$$

$$\mathcal{L}_{\text{s}} = \sum_{j \in M} \lambda_{(s,j)} \, \text{s}_j + \frac{\rho_s}{2} \sum_{j \in N} \text{s}_j^2 \tag{6}$$

$$\mathcal{C}_\rho(w; \lambda) = -\frac{1}{L} \sum_{l=1}^{L} \sum_{i \in N} p_i^w(v^{(l)}) + \mathcal{L}_{\text{rgt}} - \mathcal{L}_{\text{s}}$$

Table 3: We evaluate both RegretNet and PreferenceNet over $nxm$ auctions ("u" refers to unit-demand and "a" refers to additive), where $n$ is the number of agents and $m$ is the number of items. We measure three criteria **(a) PCA, (b) Regret Mean (STD), (c) Payment Mean (STD)**. We find that using an augmented Lagrangian approach to enforce constraints do not provide significant improvement to RegretNet, and negatively impact the performance of PreferenceNet.

| (a) | TVF | | Entropy | | Quota | |
|---|---|---|---|---|---|---|
| | RegretNet | PreferenceNet | RegretNet | PreferenceNet | RegretNet | PreferenceNet |
| 2x2 u | 100.0 | 100.0 | 100.0 | 100.0 | 100.0 | 100.0 |
| 2x4 u | 100.0 | 100.0 | 100.0 | 100.0 | 100.0 | 100.0 |
| 4x2 u | 100.0 | 100.0 | 100.0 | 81.7 | 100.0 | 100.0 |
| 4x4 u | 100.0 | 100.0 | 100.0 | 69.7 | 100.0 | 100.0 |
| 2x2 a | 100.0 | 100.0 | 100.0 | 100.0 | 100.0 | 100.0 |
| 2x4 a | 100.0 | 100.0 | 100.0 | 100.0 | 100.0 | 100.0 |
| 4x2 a | 100.0 | 100.0 | 100.0 | 100.0 | 100.0 | 100.0 |
| 4x4 a | 100.0 | 100.0 | 100.0 | 71.9 | .0 | .1 |

| (b) | TVF | | Entropy | | Quota | |
|---|---|---|---|---|---|---|
| | RegretNet | PreferenceNet | RegretNet | PreferenceNet | RegretNet | PreferenceNet |
| 2x2 u | .011 (.011) | .17 (.082) | .013 (.013) | .027 (.018) | .015 (.015) | .02 (.029) |
| 2x4 u | .01 (.008) | .136 (.086) | .008 (.008) | .061 (.057) | .012 (.017) | .904 (.185) |
| 4x2 u | .016 (.008) | .897 (.167) | .018 (.018) | .662 (.118) | .076 (.062) | .165 (.075) |
| 4x4 u | .029 (.02) | .223 (.07) | .041 (.041) | .283 (.134) | .05 (.041) | 1.317 (.303) |
| 2x2 a | .006 (.004) | .001 (.001) | .006 (.006) | .031 (.024) | .007 (.004) | .034 (.016) |
| 2x4 a | .007 (.011) | .063 (.052) | .007 (.007) | .035 (.032) | .007 (.006) | .031 (.013) |
| 4x2 a | .091 (.148) | .031 (.029) | .012 (.012) | .149 (.032) | .12 (.063) | .177 (.036) |
| 4x4 a | .363 (.323) | 0 (0) | .026 (.026) | .399 (.057) | .036 (.016) | .042 (.03) |

| (c) | TVF | | Entropy | | Quota | |
|---|---|---|---|---|---|---|
| | RegretNet | PreferenceNet | RegretNet | PreferenceNet | RegretNet | PreferenceNet |
| 2x2 u | 4.17 (.45) | 4.12 (.38) | 4.19 (.45) | 3.97 (.33) | 4.12 (.36) | 4.14 (.12) |
| 2x4 u | 4.37 (.39) | 4.33 (.3) | 4.4 (.28) | 3.86 (.15) | 4.32 (.37) | 4.95 (.26) |
| 4x2 u | 4.92 (.2) | 4.99 (.21) | 4.92 (.2) | 5.06 (.21) | 4.27 (.05) | 4.29 (.18) |
| 4x4 u | 8.78 (.39) | 8.66 (.41) | 8.78 (.42) | 6.88 (.3) | 8.69 (.39) | 8.84 (.74) |
| 2x2 a | .89 (.31) | .01 (0) | .89 (.3) | .72 (.23) | .47 (.28) | .51 (.19) |
| 2x4 a | 1.78 (.4) | 1.74 (.35) | 1.76 (.39) | .99 (.07) | 1.1 (.47) | .04 (.02) |
| 4x2 a | 1.16 (.38) | .07 (.01) | 1.17 (.22) | .2 (.04) | .34 (.05) | .18 (.04) |
| 4x4 a | 2.58 (.39) | 0 (0) | 2.22 (.3) | 1.98 (.29) | 2.51 (.32) | 1.99 (.31) |

Similarly, we also adapt the loss functions for training RegretNet to explicitly enforce the constraint using augmented Lagrangian multipliers as in [25] and observe the impact on PCA, mean regret, and mean payments. In general, adding augmented Lagrangian multipliers improves overall PCA scores for both RegretNet and PreferenceNet. However, it provides RegretNet with limited improvements in minimizing regret and maximizing payments. Moreover, augmented Lagrangian multipliers have an overall negative impact on PreferenceNet, as mean regret is higher and mean payments are lower on average. We hypothesize that since PreferenceNet enforces preferences constraints implicitly with a learned model rather than an exact signal, preference loss should not be strictly enforced with Lagrangian multipliers.

## D    Comparing Training Time between PreferenceNet and RegretNet

We compare the training time between PreferenceNet and RegretNet for several models that enforce the TVF constraint. We train each model on an unloaded machine with an RTX 2080 graphics card, and 32 GB of memory. As shown in Table 4, PreferenceNet takes nearly 50% longer to train. The training time increases proportionally with the size of the auction, the size of the MLP training set,

Table 4: We evaluate both RegretNet and PreferenceNet over $n \times m$ auctions ("u" refers to unit-demand and "a" refers to additive), where $n$ is the number of agents and $m$ is the number of items. We measure the total training time of both models and find PreferenceNet takes 50% longer to train.

| TVF | 1x2 a | 2x2 a | 2x4 a | 4x2 a | 4x4 a |
|---|---|---|---|---|---|
| RegretNet | 28m | 29m | 34m | 32m | 37m |
| PreferenceNet | 43m | 46m | 48m | 49m | 54m |

and the number of times the MLP is retrained. Carefully tuning these hyperparameters could further improve the training time of PreferenceNet.

# E   Scaling to Larger Auctions

PreferenceNet, much like RegretNet, can scale to relatively large auctions. The largest auction setting tested in the literature is the 5 agent, 10 item auction as described in [13]. Similarly, we replicate this experiment in Table 5. We find that many of the trends seen for smaller auctions still hold as the number of agents and items scale up. Surprisingly, we find that both RegretNet and PreferenceNet fail to satisfy the quota-constraint for the unit-demand setting.

We also note that as auctions get larger in the number of bidders, a baseline itemwise Myerson auction will in many settings capture a very large percentage of the possible optimal revenue, so perhaps at really huge scales, the optimal auction problem is also less interesting.

Table 5: We evaluate both RegretNet and PreferenceNet over $n \times m$ auctions ("u" refers to unit-demand and "a" refers to additive), where $n$ is the number of agents and $m$ is the number of items. We measure three criteria **(a) PCA**, **(b) Regret Mean (STD)**, **(c) Payment Mean (STD)**. We find that PreferenceNet can replicate the performance of RegretNet for larger auctions.

| (a) | TVF | | Entropy | | Quota | |
|---|---|---|---|---|---|---|
| | RegretNet | PreferenceNet | RegretNet | PreferenceNet | RegretNet | PreferenceNet |
| 5x10 u | 100.0 | 100.0 | 100.0 | 100.0 | 97.7 | 100.0 |
| 10x5 u | 100.0 | 99.8 | 100.0 | 0 | 100.0 | 100.0 |
| 5x10 a | 100.0 | 98.4 | 100.0 | 99.8 | 0 | 0 |
| 10x5 a | 100.0 | 49.8 | 100.0 | 100.0 | 0 | 0 |

| (b) | TVF | | Entropy | | Quota | |
|---|---|---|---|---|---|---|
| | RegretNet | PreferenceNet | RegretNet | PreferenceNet | RegretNet | PreferenceNet |
| 5x10 u | .108 (.072) | .082 (.035) | .051 (.051) | .044 (.021) | .093 (.036) | 2.502 (.207) |
| 10x5 u | .082 (.023) | .117 (.024) | .085 (.085) | .164 (.041) | 2.446 (.198) | 2.466 (.2) |
| 5x10 a | .142 (.3) | .054 (.036) | .293 (.293) | .504 (.141) | .521 (.371) | .517 (.383) |
| 10x5 a | .144 (.226) | .459 (.153) | .26 (.26) | .404 (.08) | .293 (.103) | .299 (.108) |

| (c) | TVF | | Entropy | | Quota | |
|---|---|---|---|---|---|---|
| | RegretNet | PreferenceNet | RegretNet | PreferenceNet | RegretNet | PreferenceNet |
| 5x10 u | 10.11 (1.22) | 11.54 (.39) | 11.43 (.49) | 11.4 (.43) | 11.52 (.4) | 12.53 (.21) |
| 10x5 u | 12.43 (.23) | 12.56 (.23) | 12.52 (.25) | 12.72 (.24) | 12.51 (.21) | 12.51 (.21) |
| 5x10 a | 6.07 (.6) | 5.63 (.46) | 5.94 (.62) | 5.64 (.53) | 6.09 (.59) | 6.09 (.59) |
| 10x5 a | 3.32 (.4) | 3.02 (.38) | 3.17 (.44) | 2.93 (.26) | 4.4 (.21) | 4.41 (.21) |

# F   Comparing PreferenceNet and RegretNet for Asymmetrical Valuations

All prior experiments were IID, where all agents sampled bids from the same valuation functions. We soften this requirement and evaluate the performance of RegretNet and PreferenceNet where all agents do not sample bids from the same distribution. Specifically, we study the auction setting with 2 agents, and 2 items. We study the case where agent 1 samples from a distribution $n$ times

Table 6: We evaluate both RegretNet and PreferenceNet over 2 agent, 2 item auctions with asymmetric input valuations. We scale the distribution for input bids of agent 1 by $n$ times the distribution for agent 2 (Denoted by $n$:1). We measure three criteria **(a) PCA**, **(b) Regret Mean (STD)**, **(c) Payment Mean (STD)**. Importantly, PreferenceNet maintains parity with RegretNet.

| (a) | TVF | | Entropy | | Quota | |
|---|---|---|---|---|---|---|
| | RegretNet | PreferenceNet | RegretNet | PreferenceNet | RegretNet | PreferenceNet |
| 2x2 2:1 | 100.0 | 100.0 | 93.5 | 99.3 | 39.2 | 100.0 |
| 2x2 4:1 | 100.0 | 100.0 | 80.0 | 98.6 | 27.3 | 100.0 |
| 2x2 8:1 | 96.5 | 100.0 | 72.9 | 96.2 | 17.1 | 100.0 |

| (b) | TVF | | Entropy | | Quota | |
|---|---|---|---|---|---|---|
| | RegretNet | PreferenceNet | RegretNet | PreferenceNet | RegretNet | PreferenceNet |
| 2x2 2:1 | .007 (.006) | .008 (.006) | .008 (.008) | .015 (.009) | .011 (.01) | .064 (.036) |
| 2x2 4:1 | .01 (.007) | .011 (.007) | .011 (.011) | .062 (.035) | .035 (.026) | .135 (.069) |
| 2x2 8:1 | .019 (.013) | .021 (.016) | .02 (.02) | .054 (.072) | .02 (.014) | .151 (.082) |

| (c) | TVF | | Entropy | | Quota | |
|---|---|---|---|---|---|---|
| | RegretNet | PreferenceNet | RegretNet | PreferenceNet | RegretNet | PreferenceNet |
| 2x2 1:2 | 1.37 (.62) | 1.36 (.61) | 1.38 (.61) | 1.39 (.61) | 1.36 (.7) | .94 (.51) |
| 2x2 1:4 | 2.44 (1.33) | 2.4 (1.34) | 2.46 (1.34) | 2.59 (1.31) | 2.58 (1.37) | 1.76 (.97) |
| 2x2 1:8 | 4.66 (2.73) | 4.57 (2.77) | 4.62 (2.8) | 4.35 (2.77) | 4.71 (2.83) | 3.07 (1.75) |

larger than agent 2, where $n = 2, 4, 8$. We find that PreferenceNet is able to satisfy the preference requirement for TVF, entropy, and quota better than RegretNet. Moreover, the average regret and payment are comparable between the two networks.

# G   Survey Protocols and Selected Responses

We now report the entirety of the two survey protocols and share select survey responses on how participants made decisions. For each survey, we also include two attention check questions similar to the provided examples to ensure that participants are actively engaged and faithfully completing the survey.

## G.1   Selected Responses

We asked participants to describe in words why they selected certain auctions as fair. We found that many of these responses can be categorized into two groups: (1) participants with clear definitions of fairness and (2) participants with vague inexpressible preferences. Below are some examples selected from among the first 100 responses in each category:

Clear Preference of Fairness

- "I considered 50/50 most fair, added the percentages both groups were off from this number and had the smaller number was the most fair."

- "I used math, subtraction, and which option had the least amount of difference"

- "I looked at each instance and figured out which case presented the least variance from 50/50"

Inexpressible Preference of Fairness

- "I just read the dialog and decided from there which one was the better choice."

- "I just clicked on what I thought was fair"

- "I went with what looked more even"

The first group described something like an explicit function for determining fairness; the second group did have preferences but could not describe them. PreferenceNet provides a mechanism to capture preferences from both types of survey participants, allowing broader participation in the auction design process.

## G.2   Measuring Preference Noise: Survey Protocol

The design of this survey is aimed at understanding how you interpret a given definition of fairness, and use this interpretation to decide if allocations of resources are fair with the given definition. You will be exploring this concept in the setting of advertising to different demographics. The survey will have two parts:

(1) familiarization with the given definition of fairness and

(2) answering questions about your understanding of the definition as applied to new scenarios.

This survey does not have correct answers. We would like you to consider each scenario and let us know what you genuinely think.

Let's begin with the familiarization process now. Consider this fairness definition.

**Fairness Definition:  An allocation is fair if the two similar demographic groups see ads at similar rates.**

Let us walk through some examples of this. Consider a company, Company A, who displays an ad.

In our first example, Company A's ad was shown to 45.0% DEMOGRAPHIC1 and 55.0% DEMOGRAPHIC2.

Considering the given definition of fairness, this allocation is **fair**.

In our second example, Company A's ad was shown to 25.0% DEMOGRAPHIC1 and 75.0% DEMOGRAPHIC2.

Considering the given definition of fairness, this allocation is **not fair**.

————————————Page Break————————————

Now answer these questions based on what you think is fair and what isn't.

————————————Page Break————————————

[The participants were shown 8 examples of this question with values X and Y randomly generated between 30 and 70.]

Question:  Company A's ad was shown to X% DEMOGRAPHIC1 and Y% DEMOGRAPHIC2. Considering the given definition of fairness, is this fair? [Yes/No]

————————————Page Break————————————

Now, consider a more complex setting where there are two companies, Company A and Company B. We are still interested in what you think is fair. Importantly, when making a determination of fairness, you must consider that the two demographic groups see both Company A and Company B's ads at similar rates.

Let us walk through some examples:

Example 1. Company A's ad was shown to 30.0% DEMOGRAPHIC1 and 70.0% DEMOGRAPHIC2. Company B's ad was shown to 30.0% DEMOGRAPHIC1 and 70.0% DEMOGRAPHIC2. Considering both companies, according to the given definition of fairness, this allocation is **fair**.

Example 2. Company A's ad was shown to 30.0% DEMOGRAPHIC1 and 70.0% DEMOGRAPHIC2. Company B's ad was shown to 30.0% DEMOGRAPHIC1 and 70.0% DEMOGRAPHIC2. Considering both companies, according to the given definition of fairness, this allocation is **not fair**.

————————————Page Break————————————

[The participants were shown 30 examples of this question with values X, Y, W, and Y randomly generated between 30 and 70.]

Question: Company A's ad was shown to X% DEMOGRAPHIC1 and Y% DEMOGRAPHIC2. Company B's ad was shown to W% DEMOGRAPHIC1 and Z% DEMOGRAPHIC2. Considering the given definition of fairness, is this fair?[Yes/No]

————————————Page Break————————————

Thanks for submitting answers to those questions.

Question: Can you describe what method or process you used to make your decisions? [Free text Response]

————————————Page Break————————————

Thank you for your time taking this survey. Please click next to return to Cint.

————————————Survey End————————————

### G.3  Group Preference Elicitation: Survey Protocol

The design of this survey is aimed at understanding how you interpret the fairness of ad placements, and how you use this interpretation to decide which allocations of resources are more fair than others. You will be exploring this concept in the setting of advertising to different demographics. The survey will have two parts:

(1) familiarization with setting and

(2) answering questions about your preferences between new scenarios.

This survey does not have correct answers. We would like you to consider each scenario and let us know what you genuinely think.

Let's begin with the familiarization process now by walking through some examples.

Consider two companies, Company A and Company B, who displays an ad.

In our first example, Company A's ad was shown to 45.0% DEMOGRAPHIC1 and 55.0% DEMO-GRAPHIC2.

In our second example, Company B's ad was shown to 25.0% DEMOGRAPHIC1 and 75.0% DEMOGRAPHIC2.

In the first part of the survey, you will be asked which of these two scenarios you think is more fair.

————————————Page Break————————————

Now answer these questions based on what you think is fair and what isn't.

————————————Page Break————————————

[The participants were shown 8 examples of this question with values X, Y, W, and Y randomly generated between 30 and 70.]

Question: Case 1: Company A's ad was shown to X% DEMOGRAPHIC1 and Y% DEMO-GRAPHIC2. Case 2: Company B's ad was shown to W% DEMOGRAPHIC1 and Z% DEMO-GRAPHIC2. Which is more fair, Case 1, Case 2? [Case 1/Case 2]

————————————Page Break————————————

Now, consider a more complex setting where you are comparing the joint placement of Companies A and B with Companies C and D. We are still interested in what you think is fair.

Let us walk through an example:

Example In Case 1: Company A's ad was shown to 30.0% DEMOGRAPHIC1 and 70.0% DE-MOGRAPHIC2. Company B's ad was shown to 30.0% DEMOGRAPHIC1 and 70.0% DEMO-GRAPHIC2.

In Case 2: Company C's ad was shown to 30.0% DEMOGRAPHIC1 and 70.0% DEMOGRAPHIC2. Company D's ad was shown to 30.0% DEMOGRAPHIC1 and 70.0% DEMOGRAPHIC2.

Considering both cases, we are going to ask you which of these two cases do you think is more fair.

——————————Page Break——————————[The participants were shown 30 examples of this question with values X1, Y1, W1, Y1, X2, Y2, W2, and Y2 randomly generated between 30 and 70.]

Question: Case 1:Company A's ad was shown to X1% DEMOGRAPHIC1 and Y1% DEMOGRAPHIC2. Company B's ad was shown to W1% DEMOGRAPHIC1 and Z1% DEMOGRAPHIC2.

Case 2: Company C's ad was shown to X2% DEMOGRAPHIC1 and Y2% DEMOGRAPHIC2. Company D's ad was shown to W2% DEMOGRAPHIC1 and Z2% DEMOGRAPHIC2.

Which is more fair, Case 1, Case 2? [Case 1/Case 2]

——————————Page Break——————————

Thanks for submitting answers to those questions.

Question: Can you describe what method or process you used to make your decisions? [Free text Response]

——————————Page Break——————————

Thank you for your time taking this survey. Please click next to return to Cint.

——————————Survey End——————————