# OpenReview forum: "PreferenceNet: Encoding Human Preferences in Auction Design with Deep Learning"
_NeurIPS.cc/2021/Conference — NeurIPS 2021 Poster_

### Official Review · Reviewer_mN2t · 2021-07-14

**Rating:** 6
**Confidence:** 3

**Summary:**

This paper studies of designing revenue optimizing auctions subject to the an additional constraint on fairness. Instead of algebraically capturing the fairness of an allocation, the authors train a neural network termed PreferenceNet that assign a fairness score to any allocation. The neural network for strategy-proof, revenue maximizing auction (RegretNet) is trained to trade-off revenue and regret with the preference loss.  The author evaluate this on small instances (2-4 bidders and 2-4 items) and contrast revenue, regret and fairness of simple RegretNet with one constrained to produce fair allocations.

One primary motivation for a deep neural network based PreferenceNet is that it can capture real world preferences from humans. To explore this aspect, the authors conducted real world studies to get user preference samples and trained a PreferenceNet on that. They also correlate preference learned from humans with mathematical models of preference.

**Limitations And Societal Impact:**

Authors do discuss limitations of their approach - in terms of human preference data gathering and model training. Particular for human preference data gathering - authors discussed the noise introduced and how they handled that. The authors also discuss the limitations of the PreferenceNet model and the situations in which it fails. One aspect lacking is looking deeper into the model trained and what it teaches about optimal auction design - this aspect is what makes the RegretNet approach interesting. Alternately the authors could discuss how this approach could scale to instances of arbitrary size.

**Main Review:**

The key idea in this paper is to train a deep neural network for scoring an allocation for fairness. This network is used to score allocations produced by RegretNet which is the deep neural network for computing revenue optimizing incentive compatible allocations. The authors provide empirical evaluation of their idea on small instances.

The paper is closely related to the paper by Kuo et al. ProportionNet: Balancing Fairness and Revenue for Auction Design with Deep Learning. The key distinction is that paper uses only TVF and the allocation is scored using the mathematical definition, while in this work the authors approximate it using a neural network.

I think this approach is interesting. The authors should explain their approach more fully and also evaluate it on a diverse set of inputs. In particular, fairness can become more restrictive if the advertisers preference for the different goods are very different. So it will be interesting to explore valuations that not identically distributed or with a larger spread of values such as exponential distributions.

Another aspect that is worth evaluating is looking into what type of auctions are designed - how is the optimal auction different with the additional fairness constraint.

Given these concerns, I believe the authors could strengthen paper a lot with a more thorough evaluation of their model.

I also have a minor clarification question. The learning process describes re-training preference net about a few rounds of training regret-net. Is that relevant even for a preference-net trained for real world data. How does that work - does it require new data gathering or some sort of heuristic based approach, what if at all would be the impact of getting that part wrong.

Minor:
- Many of the references cite the authors as author A et al. It is preferred if the authors could include the full list of authors.
- the definition of TVF is not clear. What's c, C_k, d^k(j, j')?
- please clarify that the experiment evaluation was done on bids drawn IID from U[0, 1].

Update based on author's response:
Authors provided a lot more evaluation of their approach with non-identical distributions, larger instance sizes. The authors should include this new data in the paper as it provides a more thorough evaluation of the approach. For example, we do see larger gaps emerge in the metrics with larger instances.  I had a some more questions about the results.
1) In some cases RegretNet PCA is lower than PreferenceNet, is this expected. Shouldn't RegretNet perform better?
2) Can we still say that the two approaches are similar if the PCA is very far from each other or revenue is at times more than 5-6% different?


**Time Spent Reviewing:**

3

---

> ### Author Response · Authors · 2021-08-10
> **Response to reviewer**
>
> We sincerely thank the reviewer for the helpful review. We would like to clarify some of the issues raised.
>
> The reviewer is interested in how PreferenceNet will function when trained with bids that are not IID-uniform, with larger spread. First, we confirm that all valuations in the paper are IID. Next, we examine the auction setting with 2 agents, and 2 items. We study the case where agent 1 samples from a distribution $n$ times larger than agent 2, where $n=2,4,8$. We find that PreferenceNet is able to satisfy the preference constraint for TVF, entropy, and quota better than RegretNet. Moreover, the average regret and payment are comparable between the two networks.
>
> Tables for non-symmetric distributions ((a) PCA score,(b) Regret Mean (STD),(c) Payment Mean (STD)):
>
> | **(a)**   | TVF       |               | Entropy   |               | Quota     |               |
> | :-------- | :-------: | :-----------: | :-------: | :-----------: | :-------: | :-----------: |
> |           | RegretNet | PreferenceNet | RegretNet | PreferenceNet | RegretNet | PreferenceNet |
> | 2x2 2:1   | 100\.0    | 100\.0        | 93\.5     | 99\.3         | 39\.2     | 100\.0        |
> | 2x2 4:1   | 100\.0    | 100\.0        | 80\.0     | 98\.6         | 27\.3     | 100\.0        |
> | 2x2 8:1   | 96\.5     | 100\.0        | 72\.9     | 96\.2         | 17\.1     | 100\.0        |
>
>
>
> | **(b)**   | TVF         |               | Entropy     |               | Quota       |               |
> | :-------- | :---------: | :-----------: | :---------: | :-----------: | :---------: | :-----------: |
> |           | RegretNet   | PreferenceNet | RegretNet   | PreferenceNet | RegretNet   | PreferenceNet |
> | 2x2 2:1   | .007 (.006) | .008 (.006)   | .008 (.008) | .015 (.009)   | .011 (.01)  | .064 (.036)   |
> | 2x2 4:1   | .01 (.007)  | .011 (.007)   | .011 (.011) | .062 (.035)   | .035 (.026) | .135 (.069)   |
> | 2x2 8:1   | .019 (.013) | .021 (.016)   | .02 (.02)   | .054 (.072)   | .02 (.014)  | .151 (.082)   |
>
>
> | **(c)**   | TVF          |               | Entropy      |               | Quota        |               |
> | :-------- | :----------: | :-----------: | :----------: | :-----------: | :----------: | :-----------: |
> |           | RegretNet    | PreferenceNet | RegretNet    | PreferenceNet | RegretNet    | PreferenceNet |
> | 2x2 1:2   | 1\.37 (.62)  | 1\.36 (.61)   | 1\.38 (.61)  | 1\.39 (.61)   | 1\.36 (.7)   | .94 (.51)     |
> | 2x2 1:4   | 2\.44 (1.33) | 2\.4 (1.34)   | 2\.46 (1.34) | 2\.59 (1.31)  | 2\.58 (1.37) | 1\.76 (.97)   |
> | 2x2 1:8   | 4\.66 (2.73) | 4\.57 (2.77)  | 4\.62 (2.8)  | 4\.35 (2.77)  | 4\.71 (2.83) | 3\.07 (1.75)  |
>
> The reviewer asks how the optimal auctions differ with additional fairness constraints. We -- and indeed the auction design community -- don’t know for certain what the optimal fairness-constrained auctions are even for 1 agent with explicitly known fairness constraints, although the trained auctions from this and previous work are suggestive of what they might look like. We think that this is an extremely worthwhile question, but believe it is outside the scope of this work (which is particularly concerned with capturing potentially “fuzzy” or ill-defined human preferences, such that a proof certifying optimality might not even be possible).
>
> The reviewer is interested in how we retrain PreferenceNet for human-subject experiments. Importantly, this process is no different than in the case with synthetic data.  As described in Section 4, (MLP Co-Training), we use the MLP, trained on ground truth data, to generate noisy labels, which we add back into the training set. We find that this effectively reinforces the decision boundary between positive and negative examples. Ideally, we’d like to add human labelers in the training loop to continuously sample preferences through training. If the noisy labels are incorrect, this may have a marginal negative impact on satisfying the ground truth constraint, particularly along the decision boundary of good and bad allocations.
>
> The reviewer asks us to clarify the definition of total variation fairness. The intuition for TVF is that “similar items should be treated similarly”; it is defined more clearly in Kuo et al. The $d$ is a distance between items, assumed to be known a priori, which specifies which items are similar. The more similar two items are, the less their allocations are allowed to differ. The $C_k$, for $k=1$ to $c$, defines groups of bidders who have the same definitions of similarity and so face the same fairness constraints. We will improve the clarity of this definition.
>
> The reviewer asks how our proposed method scales to auctions of arbitrary size. The largest deep-learned auction that we know of from the literature (Duetting et al.) has 5 bidders and 10 items, which still isn’t very large. We present a comparison between RegretNet and PreferenceNet for the 5 agent, 10 item auction and 10 agent 5 item auction below, but find that many of the same conclusions made in the main paper still hold.
>
> | **(a)**   | TVF       |               | Entropy   |               | Quota     |               |
> | :-------- | :-------: | :-----------: | :-------: | :-----------: | :-------: | :-----------: |
> |           | RegretNet | PreferenceNet | RegretNet | PreferenceNet | RegretNet | PreferenceNet |
> | 5x10 u    | 100\.0    | 100\.0        | 100\.0    | 100\.0        | 97\.7     | 100\.0        |
> | 10x5 u    | 100\.0    | 99\.8         | 100\.0    | 0             | 100\.0    | 100\.0        |
> | 5x10 a    | 100\.0    | 98\.4         | 100\.0    | 99\.8         | 0         | 0             |
> | 10x5 a    | 100\.0    | 49\.8         | 100\.0    | 100\.0        | 0         | 0             |
>
>
> | **(b)**   | TVF         |               | Entropy     |               | Quota         |               |
> | :-------- | :---------: | :-----------: | :---------: | :-----------: | :-----------: | :-----------: |
> |           | RegretNet   | PreferenceNet | RegretNet   | PreferenceNet | RegretNet     | PreferenceNet |
> | 5x10 u    | .108 (.072) | .082 (.035)   | .051 (.051) | .044 (.021)   | .093 (.036)   | 2\.502 (.207) |
> | 10x5 u    | .082 (.023) | .117 (.024)   | .085 (.085) | .164 (.041)   | 2\.446 (.198) | 2\.466 (.2)   |
> | 5x10 a    | .142 (.3)   | .054 (.036)   | .293 (.293) | .504 (.141)   | .521 (.371)   | .517 (.383)   |
> | 10x5 a    | .144 (.226) | .459 (.153)   | .26 (.26)   | .404 (.08)    | .293 (.103)   | .299 (.108)   |
>
> | **(c)**   | TVF           |               | Entropy      |               | Quota        |               |
> | :-------- | :-----------: | :-----------: | :----------: | :-----------: | :----------: | :-----------: |
> |           | RegretNet     | PreferenceNet | RegretNet    | PreferenceNet | RegretNet    | PreferenceNet |
> | 5x10 u    | 10\.11 (1.22) | 11\.54 (.39)  | 11\.43 (.49) | 11\.4 (.43)   | 11\.52 (.4)  | 12\.53 (.21)  |
> | 10x5 u    | 12\.43 (.23)  | 12\.56 (.23)  | 12\.52 (.25) | 12\.72 (.24)  | 12\.51 (.21) | 12\.51 (.21)  |
> | 5x10 a    | 6\.07 (.6)    | 5\.63 (.46)   | 5\.94 (.62)  | 5\.64 (.53)   | 6\.09 (.59)  | 6\.09 (.59)   |
> | 10x5 a    | 3\.32 (.4)    | 3\.02 (.38)   | 3\.17 (.44)  | 2\.93 (.26)   | 4\.4 (.21)   | 4\.41 (.21)   |

---

### Official Review · Reviewer_wcGk · 2021-07-15

**Rating:** 6
**Confidence:** 3

**Summary:**

This paper describes how we can encode socially desirable constraints in auction mechanisms learned using the regretNet framework.   The paper has two main contributions:
(1) A metric that quantifies how well these mechanisms adhere to these constraints.
(2) An neural network and a training procedure (called the preferenceNet) that can encode these constraints using exemplars of the desired allocation

The authors demonstrate the efficacy of the proposed approach by showing that it can match the performance of standard approaches on both synthetic preferences as well as human preferences

**Limitations And Societal Impact:**

Yes. (For limitations see above)

**Main Review:**

Pros:
- Unlike existing/related work, they attempt to capture socially desirable properties through data (rather than formally defining them).
- Results are quite promising - they either match or beat standard approaches (despite only being trained using weaker and noisy signals)
- Significance: This paper studies an important problem at the intersection of mechanism design and human in the loop learning
- Clarity: The paper is well written and easy to understand


Comments:
- My main concern is the inability of preferenceNet to learn the preference function exactly (which happens to play a key role in the training pipeline)
- It is unclear how $\alpha$,  $\beta$ and $\gamma$ were chosen.
- Is there an intuition behind why some constraints are harder than others?

Minor Comments:
- typo on line 193: evalate-> evaluate
- The augmented Lagrangian method is usually the sum of squares of the penalty term (rather than the square of the sum of penalty terms)
 $$\frac{\rho_r}{2}\left(\sum_{i} rgt_i(w)\right)^2 => \frac{\rho_r}{2} \sum_{i} \left(rgt_i(w)\right)^2$$


**Time Spent Reviewing:**

3 hours

---

> ### Author Response · Authors · 2021-08-10
> **Response to reviewer**
>
> We sincerely thank the reviewer for their thorough review. Our response to their questions follows.
> The reviewer is concerned that PreferenceNet is unable to learn preferences exactly. However, through experimental evaluation we find that by learning an approximation of the exact preference function, we are still able to satisfy the preference (as measured by PCA) effectively.
>
> We believe people themselves don’t know their preferences exactly, but we can still capture their ideas well (Figure 5). We study the impact of noise (which shows that we are robust to not being able to exactly model the preference) in the supplement (Appendix A) and find that we are generally robust to extreme perturbations.
>
> The reviewer asks how we select the parameters alpha, beta, and gamma. We select these hyperparameters by tuning on a held-out validation set to minimize the performance difference between RegretNet and PreferenceNet. We will make this more clear.
>
> The reviewer asks if we have intuition of why some constraints are harder to learn. Based on experimental evidence, we find that constraints are harder to learn when there are fewer potential allocations that satisfy the constraint (i.e. few positives and many negative examples). For example, training a model to satisfy a quota of 49% in a 2 agent, 1 item system is much harder than satisfying a quota of 25% in a 2 agent, 1 item system. Additionally, constraints that conflict with the optimal auction are more difficult to learn since the penalty for violating regret (enforced by a lagrangian multiplier) is much greater than violating the preference constraint.
>
> We thank the reviewer for spotting the typo with the Lagrangian. We have corrected this.

---

### Official Review · Reviewer_fFgo · 2021-07-16

**Rating:** 5
**Confidence:** 3

**Summary:**

This paper proposes PreferenceNet, which is a new deep learning approach for designing revenue-maximizing auctions that encodes human preferences. PreferenceNet extends RegretNet [Duetting et al. 2019] with modified loss functions and additional constraints.

**Limitations And Societal Impact:**

The authors have addressed the limitations and potential negative societal impact of this work.

**Main Review:**

Overall I find the paper well-written and clear to follow.  The PreferenceNet proposed by the authors extends from RegretNet, with additional MLP layers that are trained to capture implicit human preferences, modified loss functions and constraints. The authors also conducted  two surveys to further study human preferences for auction outcomes, and I find the conduction of the surveys useful and important, as they provide actual feedbacks from human participants. Below are a few questions and concerns I have.

First is about the scalability and flexibility of PreferenceNet. In the current formulation, the ground truth labeling function based on the underlying preference ( s(b) ) is constrained to be a binary label, and the input is a whole vector of allocations for the given bid. However, in practice the decision ("good" or "bad") could be hard to provided by human, especially when the number of items is large. For example, human could be bad at deciding if two vectors of allocations are actually similar or not. This can make the preference feedbacks obtained very noisy. I would also be curious to see more details on experimentally how much additional computation PreferenceNet requires compared to the RegretNet.

Second is, the deep learning based approach estimates violations of the strategyproofness and fairness constraints using a gradient-based method which is effective during training, but may not accurately reflect the true extent to which strategyproofness or fairness constraints are violated. Therefore I find the design of the loss functions and constraints for PreferenceNet intuitive, but it is unclear theoretically how these affect the generalization bound for strategyproofness. Besides, will the fairness constraints render the optimal auction infeasible in some cases? Is it possible that each agent has incompatible preferences, and PreferenceNet cannot satisfy the overall preferences for all agents? I would be interested in seeing more elaboration on this side.

typos:
Eq(3) c, C_k undefined
Eq(4) numerator missing index

**Time Spent Reviewing:**

7

---

> ### Author Response · Authors · 2021-08-10
> **Response to reviewer**
>
> We sincerely thank the reviewer for their helpful review. We would like to clarify some of the issues raised.
>
> The reviewer correctly identities that eliciting preferences from humans is difficult, especially for large auctions. Therefore, to reduce the overall noise in assigning preference labels, we aggregate the preferences of the group (as shown in Figure 2). In practice, we find that using this pairwise comparison strategy, the decisions of the group (despite the noisy preferences of individuals) have a clear decision boundary between good and bad allocations (as shown in Figure 4).
>
> The reviewer asks for a comparison between the training time of RegretNet and PreferenceNet. The training time of PreferenceNet increases with the number of training examples used to train the MLP, and increases as the size of the auction increases (like RegretNet). We present these results below. We train all models in this comparison using 1 RTX 2080 GPU and 32 GB RAM. We only focus on additive valuation auctions, and the total variation fairness constraint as these do not dramatically alter the result. We find that training time increases proportionally with the size of the auction, the size of the MLP training set, and the number of times the MLP is retrained. Carefully tuning hyperparameters will further improve the training time of PreferenceNet.
>
> | TVF           | 1x2 a | 2x2 a | 2x4 a | 4x2 a | 4x4 a |
> | :------------ | :---: | :---: | :---: | :---: | :---: |
> | RegretNet     | 28m   | 29m   | 34m   | 32m   | 37m   |
> | PreferenceNet | 43m   | 46m   | 48m   | 49m   | 54m   |
>
> The reviewer asks how PreferenceNet impacts the generalization bounds for strategyproofness. Once the mechanism is actually trained using the preference loss, it is just another function mapping bids to allocations and payments. Therefore, the existing generalization bounds for regret and revenue (which in Duetting et al. say that with high probability, empirical test-time regret/revenue should be a good estimate of the true expected value) should still hold.
>
> Next, the reviewer correctly states that “deep learning based approach estimates violations of the strategyproofness and fairness constraints using a gradient-based method which is effective during training, but may not accurately reflect the true extent to which strategyproofness or fairness constraints are violated.” While we agree with this statement, we believe that this is out of scope for the current paper. Curry et. al. addresses similar issues with RegretNet, and can potentially be adapted to PreferenceNet in future work.
>
> The reviewer asks if fairness constraints render optimal auctions infeasible in some cases. We believe that this is true in general, not just limited to PreferenceNet. For instance, even in the 1 agent, 2 item, IID-uniform case, there are regions where the optimal auction allocates item 1 with probability 1, and item 2 with probability 0. Under most constraints we consider, this would not be allowed, so the optimal auction would have to change.
>
> Lastly, the reviewer asks how PreferenceNet will satisfy incompatible preferences among survey respondents. We ask many survey respondents the same question and choose to use their majority vote on a given example as a simple solution to this problem. There might be better ways of aggregating incompatible preferences, but we believe that this type of social choice issue is outside the scope of the paper. Since we use the majority vote of all agents for a particular example to determine the preference label $s(b)$, we satisfy the majority of agents. We study this in the supplemental (Appendix B) and find that at least with different definitions of fairness, most agents’ preferences are satisfied.

---

### Official Review · Reviewer_hyQW · 2021-07-16

**Rating:** 7
**Confidence:** 2

**Summary:**

This paper covers a somewhat interesting and novel problem: learning constraints of an auction setting, and then learning to run an auction that respects those. Fairness in allocations is cited as a justification for this in practice, where it may be beneficial to learn from a human’s choices to determine how to decide who should win in the auction, for example to determine similar allocations across demographics.

The approach builds on the RegretNet line of work that learns to run revenue maximizing auctions. The approach broadly is to learn a feasibility constraint that is scored by human subjects, and then use that in the RegretNet architecture to learn to generate auctions that respect that constraint. In this work, the motivating constraint is a binary classification of fairness. The implementation is a composition of learning then constraint, followed by leveraging RegretNet to learn according to that function.

To support the approach, the authors first compare PreferenceNet to RegretNet trained with the exact constraints, and see similar performance suggesting some amount of usefulness. The authors then train PreferenceNet on responses from human subjects in two surveys regarding the binary qualitative fairness of allocations.


**Ethical Concerns:**

(addressed above)

**Limitations And Societal Impact:**

The work has one primary potential for negative societal impact: human responses should not be expected to be unbiased. The authors do address this, though perhaps a stronger statement regarding where such approaches would be deemed to be appropriate would be helpful.

**Main Review:**

The paper delivers on a novel use for machine learning in auctions. The significance of the approach remains a little open to interpretation: I don’t know that fairness is the constraint that makes the most sense for this approach, but there is no reason for the approach to be limited to fairness as the measure used - in such cases is it better to have the biases explicit or implicit as a result of human responses? In some ways it seems that the decision of what is fair is moved from the explicit constraint to the survey question, either explicitly here or implicitly in any use of this approach.

The paper is generally well written and comprehensive in exploring the performance of PreferenceNet in the fairness area, both comparing to a baseline RegretNet with perfect knowledge of the constraint and then examples trained based on human input.

Evaluating the architecture choices is outside of my area of expertise.

Small notes:
Fig 1: doesn't seem to be particularly helpful as a comparison, or I am missing the intuition that would give this a half page here.
Line 193: eval[u]ate

**Time Spent Reviewing:**

5

---

> ### Author Response · Authors · 2021-08-10
> **Response to reviewer**
>
> We sincerely thank the reviewer for their thorough review. Our response to their questions follows.
>
> The reviewer suggests that our proposed method is broadly applicable outside of enforcing fairness constraints for auction allocation. We agree with this point -- other types of allocation preferences could be captured using the same strategy (including notions of diversity, which we do consider).
>
> The reviewer correctly points out the potential for encoding bias (either implicitly through data, or explicitly through a mathematical formulation) in the auction results. On the one hand, if the stakeholders have preferences that are biased, then the results will also follow this bias. Our approach might make complex or flawed definitions harder to understand and critique, while explicit formulas might be open to critique. However, current methods which require explicit definitions limit the number and type of socially desirable auction designs. Our method is suited for situations where trustworthy stakeholders may not be able to write down their preferences as a formal definition but nevertheless have some preferences which should be captured. Given these tradeoffs, we believe the idea of capturing implicit human preferences is still valuable and is sometimes the right choice.
>
> Lastly, we wish to clarify the purpose of Figure 1. First, we show that RegretNet and PreferenceNet have identical performance when measured using both standard qualitative metrics (allocation graph) and our proposed quantitative metric (PCA), indicating an equivalency between the metrics. Next, we use the allocation graph to motivate the limitations of existing qualitative evaluation for analyzing larger auctions. We will shrink the figure to create more space for the above discussion.

---

### Decision · Program_Chairs · 2021-09-27

**Decision:**

Accept (Poster)

**Comment:**

The review team sentiment is positive about the paper: it is a new dimension to the problem of automated auction design via NN that makes it more practical and allows us to capture subtle notions like the perception of fairness? There were initial concerns around scalability and experiments on harder (non-iid) datasets, but those were adequately addressed by the authors in the discussion. Please incorporate those in the final version.

There are additional concerns on whether biases could be introduced in the model and that through this methodology biases would be harder to identify/correct since they are not explicitly written in a formula, but hidden in the data. While I share the concern of the reviewers, I think this is inherent from this type of approach, so we need to live with that.

Finally, our recommendation is to accept the paper.